# Deformable Image Registration uncertainty quantification using deep learning for dose accumulation in adaptive proton therapy

Smolders A.[1,2], Lomax T.[1,2], Weber DC.[1], and Albertini F.[1]

[1] Paul Scherrer Institute, Center for Proton Therapy, Switzerland
[2] Department of Physics, ETH Zurich, Switzerland

**Abstract.** Deformable image registration (DIR) is a key element in adaptive radiotherapy (AR) to include anatomical modifications in the adaptive planning. In AR, daily 3D images are acquired and DIR can be used for structure propagation and to deform the daily dose to a reference anatomy. Quantifying the uncertainty associated with DIR is essential. Here, a probabilistic unsupervised deep learning method is presented to predict the variance of a given deformable vector field (DVF). It is shown that the proposed method can predict the uncertainty associated with various conventional DIR algorithms for breathing deformation in the lung. In addition, we show that the uncertainty prediction is accurate also for DIR algorithms not used during the training. Finally, we demonstrate how the resulting DVFs can be used to estimate the dosimetric uncertainty arising from dose deformation.

**Keywords:** Deformable image registration · Proton Therapy · Adaptive Planning · Uncertainty · Deep Learning

## 1 Introduction

Due to their peaked depth-dose profile, protons deposit a substantially lower dose to the normal tissue than photons for a given target dose [19]. However, the location of the dose peak is highly dependent on the tissue densities along the beam path, which are subject to anatomical changes throughout the treatment. Target margins are therefore applied, reducing the advantage of proton therapy (PT) [19]. The need to account for anatomical uncertainties can be alleviated using daily adaptive PT (DAPT), where treatment is reoptimized based on a daily patient image [1]. DAPT yields a series of dose maps, each specific to a daily anatomy. One important step of DAPT is to rely on the accurate accumulation of these doses for quality assurance (QA) of the delivered treatment and to trigger further adaptation [13, 12, 7, 3]. To this end, the daily scans are registered to a reference and their corresponding doses are deformed before summation. In the presence of deforming anatomy, deformable image registration (DIR) is used [7, 22, 25]. However, DIR is ill-posed [4], which results in dosimetric uncertainty

after deformation. Substantial work has been performed to quantify this, summarized in [7], but there remains a clear need for methods predicting uncertainty associated with DIR and its effect on dose deformation [4, 18].

In this work, an unsupervised deep learning (DL) method is presented to predict the uncertainty associated with a DIR result. Section 2 describes our method. The results of hyperparameter tuning on the predicted registration uncertainty are presented in section 3, followed by the effect on the dosimetric uncertainty arising from dose deformation. Section 4 provides a discussion and conclusions are stated in section 5.

## 2 Methods

Our work aims to estimate the uncertainty of the solution of an existing DIR algorithm. It is based upon a probabilistic unsupervised deep neural network for DIR called VoxelMorph [8]. The main equations from [8] are first summarized, after which the changes are described.

### 2.1 Probabilistic VoxelMorph

With $f$ and $m$ respectively a fixed and a moving 3D volume, here CT images, a neural network learns $z$, the latent variable for a parameterized representation of a deformable vector field (DVF) $\Phi_z$. The network aims to estimate the conditional probability $p(z|f,m)$, by assuming a prior probability $p(z) = \mathcal{N}(0, \Sigma_z)$, with $\Sigma_z^{-1} = \Lambda_z = \lambda(D - A)$, $\lambda$ a hyperparameter, $D$ the graph degree matrix and $A$ the adjacency matrix. Further, $f$ is assumed to be a noisy observation of the warped moving image with noise level $\sigma_I^2$, $p(f|m,z) = \mathcal{N}(m \circ \Phi_z, \sigma_I^2 I)$. With these assumptions, calculation of $p(z|f,m)$ is intractable. Instead, $p(z|f,m)$ is modelled as a multivariate Gaussian

$$q_\Psi(z|f,m) = \mathcal{N}(\mu_{z|f,m}, \Sigma_{z|f,m}) \tag{1}$$

with $\Psi$ the parameters of the network which predicts $\mu_{z|f,m}$ and $\Sigma_{z|f,m}$ (Fig. 1). The parameters $\Psi$ are optimized by minimizing the KL divergence between $p(z|f,m)$ and $q_\Psi(z|f,m)$, yielding, for $K$ samples $z_k \sim q_\Psi(z|f,m)$, a loss function

$$\mathcal{L}(\Psi, f, m) = \frac{1}{2\sigma_I^2 K} \sum_k ||f - m \circ \Phi_z||^2 + \frac{\lambda}{4} \sum_{i=1}^{m} \sum_{j \in N(i)} (\mu_i - \mu_j)^2$$
$$+ tr(\frac{\lambda}{2}(D - A)\Sigma_{z|f,m}) - \frac{1}{2}log(|\Sigma_{z|f,m}|) + cte \tag{2}$$

with $N(i)$ the neighboring voxels of voxel $i$. When $\Sigma_{z|f,m}$ is diagonal, the last two terms of Eq. 2 reduce to $\frac{1}{2}tr(\lambda D\Sigma_{z|f,m} - log(\Sigma_{z|f,m}))$.

## 2.2 Combining deep learning with existing DIR software

Because the performance of DL based DIR is generally below conventional methods [9, 10, 24], our network aims to predict the uncertainty associated with a DVF generated by another algorithm without predicting the DVF itself. We therefore extend the VoxelMorph architecture to include the output DVF of an existing DIR algorithm (Fig. 1). First, an existing algorithm is ran on $f$ and $m$, after which the resulting DVF is concatenated to $f$ and $m$ as network input. The network only predicts a diagonal matrix $G$, which is used to calculate $\Sigma_{z|f,m}$ (see Sec. 2.3), and the mean field $\mu_{z|f,m}$ is taken as the output of the DIR algorithm.

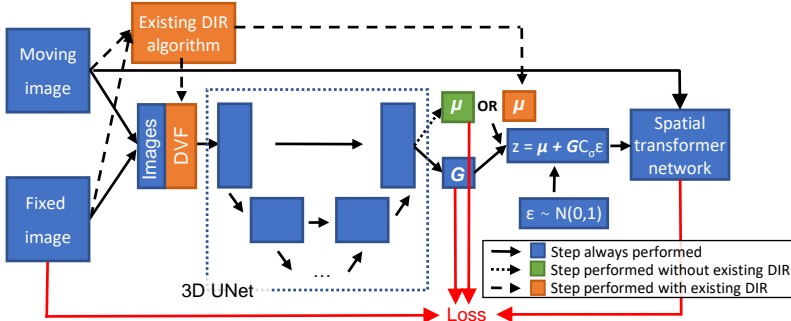

Fig. 1: Schematic network architecture. In case an existing DIR method is used, the resulting DVF of this algorithm is concatenated with the fixed and moving image, resulting in a 5xHxWxD tensor as network input. A 3D UNet predicts a diagonal matrix $G$, and taking the DVF of the existing DIR as mean field $\mu$, DVF samples are generated with the reparametrization trick as $z = \mu + GC_{\sigma_c}\epsilon$ (see Sec. 2.3) [15]. Contrarily if no existing DIR is used, the network only receives the fixed and moving image as input and predicts a mean DVF besides G.

## 2.3 Non-diagonal covariance matrix

Dosimetric uncertainty will be estimated by sampling $q_\Psi(z|f, m)$, requiring spatially smooth samples. Nearby vectors can be correlated with a non-diagonal covariance matrix. However, a full covariance matrix cannot be stored in memory because it would require storing $(3 \times H \times W \times D)^2$ entries, which for a 32 bit image of $256 \times 265 \times 96$ requires 633 TB, compared to 25 MB for the diagonal elements. In [8] a non-diagonal $\Sigma_{z|f,m}$ is proposed by Gaussian smoothing of a diagonal matrix $G$, i.e. $\Sigma_{z|f,m} = C_{\sigma_c}GG^TC_{\sigma_c}^T$, but it is shown that this is unnecessary because the implemented diffeomorphic integration smooths the samples sufficiently. Because the existing DIR solutions are not necessarily diffeomorphic, we do not apply integration, which implies the need for a non-diagonal $\Sigma_{z|f,m}$.

Similar to [8], we apply Gaussian smoothing but invert the order $\Sigma_{z|f,m} = GC_{\sigma_c}C_{\sigma_c}^TG^T$ which yields a fixed correlation matrix $\rho = C_{\sigma_c}C_{\sigma_c}^T$. This has the

advantage that the variance of the vector magnitude at voxel $i$ is only dependent on the corresponding diagonal element $G_{i,i}$ and not on its neighbors. Furthermore, it allows to simplify the calculation of the loss terms in Eq. 2. Rewriting the last two terms of Eq. 2 with $\Sigma_{z|f,m} = GC_{\sigma_c}C_{\sigma_c}^T G^T$ results in

$$
\begin{aligned}
&tr(\frac{\lambda}{2}(D-A)\Sigma_{z|f,m}) - \frac{1}{2}log(|\Sigma_{z|f,m}|) \\
&= \sum_{i=1}^{m}\sum_{j=1}^{m}(\frac{\lambda}{2}(D-A)_{i,j}(\Sigma_{z|f,m})_{i,j}) - \frac{1}{2}log(|GG^T|) - \frac{1}{2}log(|C_{\sigma_c}C_{\sigma_c}^T|)
\end{aligned}
\tag{3}
$$

with $i$ and $j$ respectively the row and column indices, $m$ the number voxels and $log(|C_{\sigma_c}C_{\sigma_c}^T|)$ a constant which can be excluded from the loss function. For each row (or voxel) $i$, the matrix $(D-A)$ has only 7 non-zero elements (the voxel itself and its 6 neighboring voxels), so that only the corresponding 7 elements in $\Sigma_{z|f,m}$ are needed to evaluate the loss function. By precomputing the 7 corresponding elements of $\rho = C_{\sigma_c}C_{\sigma_c}^T$, the first term of Eq. 3 becomes

$$
\sum_{i=1}^{m}\sum_{j=1}^{m}(\frac{\lambda}{2}(D-A)_{i,j}(\Sigma_{z|f,m})_{i,j}) = \sum_{i=1}^{m}\sum_{j\in N(i)}(\frac{\lambda}{2}(D-A)_{i,j}\rho_{i,j}G_{i,i}G_{j,j})
\tag{4}
$$

with $N(i)$ the neighbors of voxel $i$, which allows fast evaluation of $\mathcal{L}$ without the need of storing large matrices.

### 2.4   Training

52 CT scan pairs from 40 different patients with various indications treated at the Centre for Proton Therapy (CPT) in Switzerland are used for training. The pairs consist of one planning and one replanning or control CT from a proton treatment, and are therefore representative of both daily and progressive anatomical variations in DAPT. Scans are rigidly registered using the `Elastix` toolbox [16] and resampled to a fixed resolution $1.95 \times 1.95 \times 2.00$ mm, most frequently occurring in the dataset. The Hounsfield units are normalized with $\frac{HU+1000}{4000}$. Patches with a fixed size $256 \times 256 \times 96$ are randomly cropped from the full CTs during training and axis aligned flipping is applied as data augmentation.

The network is implemented in `Pytorch` [20] and training is ran on GPUs with 11 GB VRAM. A 3D UNet is used [8] with an initial convolution creating 16 feature maps, which are doubled in each of the 3 consecutive downsampling steps. The features are upsampled 3 times to their original resolution. The parameters are optimized with `Adam` [14] with initial learning rate $2 \cdot 10^{-4}$, which is halved 6 times during 500 epochs. Gaussian smoothing of the diagonal covariance matrix has a fixed kernel size of 61 voxels and blur $\sigma_c = 15$.

We train networks to predict the uncertainty associated with three existing DIR algorithms: a b-spline and a demon implementation in Plastimatch and a non-diffeomorphic VoxelMorph predicting both $\mu_{z|f,m}$ and $\Sigma_{z|f,m}$. The parameters for b-spline and demon are taken from [17, 2]. Furthermore, we verify whether these networks can be used to predict the uncertainty of other DIR algorithms by evaluating them on the results of a commercial DIR in Velocity.

## 2.5   Validation

The hyperparameters $\lambda$ and $\sigma_I^2$ are tuned for each method by quantitatively evaluating the predicted uncertainty on the publicly available 4DCT DIRLAB lung deformation dataset [6, 5]. It contains 10 CT scan pairs with each 300 annotated landmarks (LM). These scans are split equally in a validation and test set. We maximize the probability of observing the moving landmarks $\vec{x}_m$ given the predicted probabilistic vector field, which, for a given set of CTs, is calculated as

$$p(LMs) = \prod_i^{CTs} \prod_j^{LM} p(\vec{x}_{m,i,j}|DVF_i), \tag{5}$$

assuming for simplicity that each landmark is independent of the others, which is reasonable if the landmarks are sufficiently far apart. Note that the probability of observing exactly $\vec{x}_m$ is infinitesimally small because the variables are continuous. We therefore maximize the probability that $\vec{x}_m$ is observed within a cube of 1 mm$^3$ around it with a homogeneous probability density, which is the same as maximizing the probability density at $\vec{x}_m$. We discard the 1% least probable points because $p(LMs)$ is heavily affected by the outliers due to the extremely low probability density at the tails of a normal distribution. Furthermore, we maximize the mean log $p(LMs)$ to avoid that the absolute value is dependent on the number of landmarks.

## 3   Results

### 3.1   Hyper parameter tuning

The optimal hyperparameters are $\lambda = 10$ and $\sigma_I^2 = 10^{-4}$ for both b-spline and demon (Fig. 2). Further, using both the networks trained on demon and b-spline, we find that the the network trained with b-spline and $\lambda = 5$ and $\sigma_I^2 = 10^{-4}$ yields the highest average log $p(LMs)$ for Velocity (not shown).

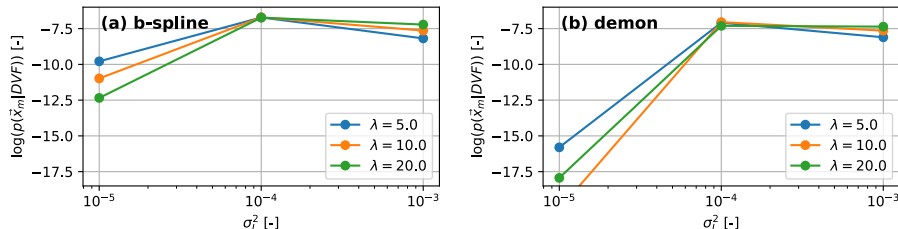

Fig. 2: Average log probability of observing moving landmarks $\vec{x}_m$ of the validation set for varying values of $\sigma_I^2$ and $\lambda$ including an existing DIR output. Similar results were found for the test set (not shown).

For VoxelMorph, the hyperparameters influence both $\mu_{z|f,m}$ and $\Sigma_{z|f,m}$. Eq. 2 shows that the trade off between similarity and smoothness is determined by

the product $\lambda\sigma_I^2$. Therefore, we first minimize the target registration error (TRE) on the validation set by varying $\lambda\sigma_I^2$ (keeping $\lambda = 2$), which yields a minimum TRE around $\lambda\sigma_I^2 = 2 \cdot 10^{-3}$. Varying $\lambda$ and $\sigma_I^2$ while keeping $\lambda\sigma_I^2 = 2 \cdot 10^{-3}$ results in a maximum $p(LMs)$ for $\sigma_I^2 = 5 \cdot 10^{-4}$. $p(LMs)$ is however lower than for the networks including the conventional (i.e non deep learning) DIRs, indicating that these methods predict better probability distributions.

Fig. 3 shows the uncertainty prediction for a lung CT in the DIRLAB dataset. As expected, the predicted uncertainty is low in regions with high contrast and high where contrast is low. Further, the Jacobian determinant is $< 0$ for on average $0.01\%$ of the voxels in sampled DVFs for the DIRLAB dataset, which, together with visual inspection, indicates that samples are sufficiently smooth.

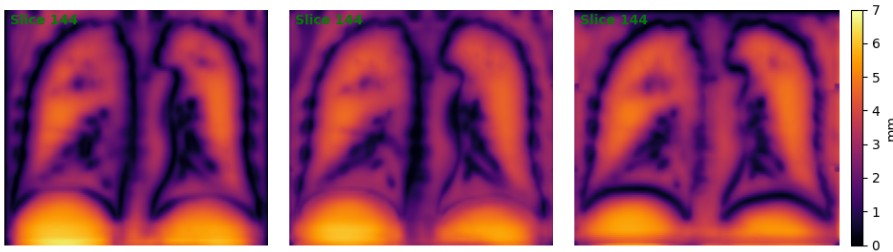

Fig. 3: Predicted uncertainty $\sigma_p$, i.e. the square root of the diagonal elements of $\Sigma_{z|f,m}$, in the sagittal (left), coronal (middle) and axial (right) direction for one example patient in the test set.

Comparing the target errors and their predictions for the tuned networks for all DIRLAB scans yields several conclusions (Fig. 4). First of all, our method is able to fairly accurately predict the uncertainty of multiple existing DIR algorithms. Secondly, the error prediction of Velocity shows that it is possible to predict the error from a DIR algorithm even if it was not used to train the network. Lastly, the average error is higher and the uncertainty prediction is worse for VoxelMorph than for the existing DIR algorithms, as expected from [9, 10, 24]. However, the performance can likely be improved by diffeomorphic integration, network adjustments or using more data, but this is not within the scope of the current study.

### 3.2   Dose deformation

We create probabilistic dose maps by sampling the probabilistic DVF and warping the dose with the different samples. We focus here on the result of a single deformation to highlight the dosimetric uncertainty associated with warping. Even though the predicted DVFs have assumed to be Gaussian, the probabilistic dose maps are not. We therefore keep the individual samples and use a finite-sample distribution to approximate the probabilistic dose map.

The dose received by the tumor and organs at risk (OARs) is in PT frequently evaluated with dose volume histograms (DVHs). Probabilistic DVHs can be constructed from the probabilistic dose map. Here, the lower and upper

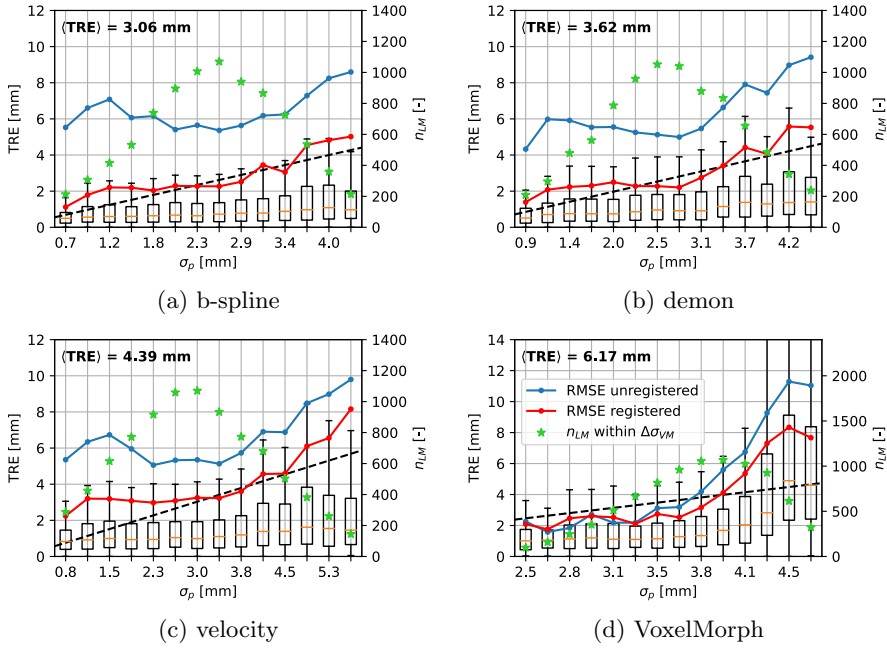

Fig. 4: TRE as a function of the predicted uncertainty $\sigma_p$ for all DIRLAB scans. For each subplot, $\sigma_p$ is divided into 15 equal intervals and the distribution of the TREs within each interval is plotted as a box, together with the unregistered and registered root mean squared error (RMSE). The number of landmarks $n_{LM}$ within each interval is also shown (right axes). If the TREs were normally distributed and the networks had a perfect prediction, the registered RMSE would be exactly equal to the predicted $\sigma_p$ (dashed line).

bound of the DVH depict for each volume increment respectively the 5th and 95th percentile of all sampled doses (Fig. 5).

Verifying whether the dosimetric uncertainty is realistic is non-trivial. Previous work [17, 2] quantified it by warping the dose with several DIR algorithms and calculating the dose differences between the results. Similarly, here we verify whether the warped dose with three conventional DIR algorithms falls in between our predicted lower and upper bound (Fig. 5). Using the same dataset of 7 lung cancer patients with each 9 repeated CTs as in [17, 2], we find that the dose in on average 97% of the volume of the OARs (heart, esophagus and medulla) lies between the bounds predicted for b-spline. For the planning target volume (PTV) and gross tumor volume (GTV) it is on average 81%.

## 4 Discussion

Despite the promising preliminary results, more work is required before the method can be used in the clinic. Our approach should be verified on a dataset

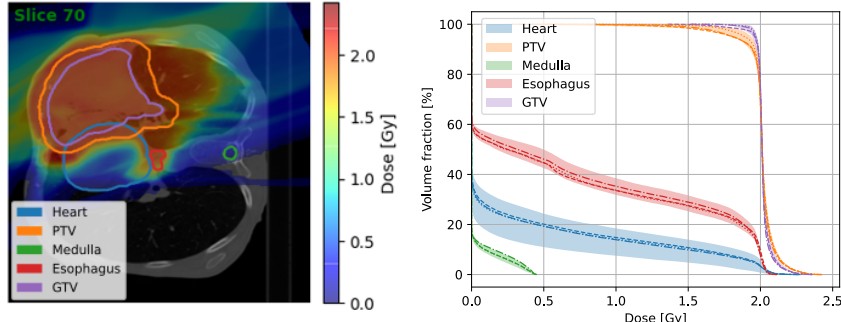

Fig. 5: Left: example of a deformed dose map with b-spline, overlayed with contours of the gross tumor volume (GTV), planning target volume (PTV) and three OARs. Right: corresponding probabilistic DVH as calculated with the optimal network for b-spline (shaded area). The dashed, dotted and dash-dotted lines represent the DVH for warped doses with three commercial DIR softwares, respectively Mirada, Raystation Anaconda and Velocity.

including typical deformations that occur during the course of six weeks of treatment, and not only during one breathing cycle. To that end, a dataset with typical anatomical deformations is currently being landmarked at the CPT.

Even for the dataset under study, the error prediction is clearly not perfect. This can be due to several factors, among which imperfect annotation, lack of training data or inaccurate model assumptions. One important assumption is the Gaussian vector field. Although our results show that it is not unreasonable to assume that the errors are Gaussian, further research should look whether other probability distributions yield better results. Unfortunately, other analytical distributions are often mathematically more complex making exact treatment as in Eqs. 2 and 3 difficult. Learning a discretized posterior could resolve this [11, 10, 21, 23].

The trained networks capture most of the dosimetric variations found in the OARs when running conventional DIRs. By contrast, for the GTV and PTV only 81% of the doses lie between the error bars, significantly below the expected 90% given the 5th and 95th percentile error bounds. However, we found that this value increases to 91% by simply adding a small margin to the error bounds (i.e. by increasing the upper and decreasing lower bound by only 0.1% of the dose). This indicates that the deviation from the error bounds is mostly very small.

## 5    Conclusion

In this work, a probabilistic unsupervised deep learning method for deformable image registration is presented to predict the uncertainty associated with DIR solutions. It is shown that the method can accurately predict the uncertainty of various conventional DIR algorithms and that the combination of deep learning with conventional DIR yields superior results than using deep learning alone.

## Acknowledgments

This work has received funding from the European Union's Horizon 2020 Marie Skłodowska-Curie Actions under Grant Agreement No. 955956

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
