# OpenReview forum: "Deformable Image Registration uncertainty quantification using deep learning for dose accumulation in adaptive proton therapy"
_WBIR.info/2022/Workshop/Biomedical_Imaging_Registration — WBIR 2022_

### Official Review · Reviewer_ZzDo · 2022-02-14

**Rating:** 2
**Confidence:** 4

**Deanonymize Review:**

no

**Detailed Comments:**

1. The motivation for the approach is unclear. In particular, it is unclear why the proposed system should be expected to result in valid uncertainty estimates for other registration algorithms and why it would be meaningful to combine other approaches with the Voxelmorph approach in this form. I assume this also would mean that at test time one needs to provide the standard registration as input.
2. The technical novelty appears to be low (i.e., adding the output of another algorithm to the input of the network and slightly changing the smoothing of the covariance matrix).
3. Besides concatenating the DVF of an existing algorithm to the input in Fig. 1, it is unclear what the second output (which appears to connect to \mu) does.
4. It is not clear how the uncertainty estimates are validated as Fig. 4 shows large discrepancies to the dashed line (which seems to be the expected optimal uncertainty).
5. Based on the presented results, it appears that the method is not working reliably. Specifically, the predicted lower and upper bounds seem to be violated in 14% of the cases of the organs at risk and in 45% of the cases for the planning target volume and gross tumor volume.


**Paper Type:**

both

**Strengths Weaknesses:**

This work explores uncertainty prediction of registration algorithms in the context of adaptive proton therapy. In particular, the work heavily builds upon the probabilistic Voxelmorph formulation developed by Dalca et al. The main differences to this prior work are that the approach 1) is not diffeomorphic, but instead of using a stationary velocity field, directly outputs displacements; 2) uses Gaussian smoothing of the covariance matrix of the latent vector of the variational approximation (but instead of the smoothing proposed by Dalca et al. swaps the order of the smoothing operation), and 3) adds the output of other registration algorithms (Demons, Bspline, and a commercial algorithm in Velocity) to the input of the Voxelmorph network.

The goal of the project of improving uncertainty estimates for better computations of dose accumulation is significant. However, there are several technical concerns with regards to the motivation of the approach and its validation as summarized in the detailed comments below.

---

### Official Review · Reviewer_wCP9 · 2022-02-19

**Rating:** 4
**Confidence:** 4

**Deanonymize Review:**

no

**Detailed Comments:**

(1) Fig.1 shows that the uncertainty map is predicted with a 3D-UNET. Although the backbone is borrowed from the probabilistic VM, the main idea of the VAE embedded in the VM has been totally For example, this work directly used a whole 3D-UNET to predict the uncertainty map instead of using the encode path. Then, (I) how to obtain the variance? (II) Is it reasonable to sample the v ariance from a fixed correlation matrix? (III) how to link the mean feature of the images and the mean deformations to the corresponding uncertainty of a deformation? (IV) why can the initial velocity represent the final deformation field even if the network takes DF from conventional methods as one input?

(2) (I)Please be careful with the sentence like “Because the existing DIR solutions are not necessarily diffeomorphic” corrects in expression. In fact, the diffeomorphic registration is an ideal status and is always the goal of registration. If the assumption is wrong, the entire method would be incorrect.

(II) Meanwhile, the procedure of simplifying the covariance matrix to the non-diagonal matrix is unclear.

(III) “For each voxel, the matrix (D-A) has only 7 non-zero elements, so that only corresponding 7 elements in sigma(z|f,x)”, why only seven non-zero elements?

(3) The VM has to predict both the deformation and the uncertainty map, while the conventional methods only have to predict the deformation fields, so the comparison is unfair for the learning-based method.

**Paper Type:**

both

**Strengths Weaknesses:**

This work attempts to predict the uncertainty of deformation via a probabilistic VM architecture to f acilitate adaptive radiotherapy. But many details in the proposed methodology are unclear. Especially, the network lacks convincing in clearly describing the relationship between the output uncertainty maps and the inputs (image pair + any initial DDF from conventional methods). Overall, this work is not easy to follow and not well prepared.

---

### Official Review · Reviewer_iCyQ · 2022-02-19

**Rating:** 3
**Confidence:** 4
**Recommendation:** Short Oral

**Deanonymize Review:**

no

**Detailed Comments:**

Minor:
Figure 2 should ideally have similar y-axis for the two comparison methods.
TRE is not defined.
Are the references provided in the correct format? It’s a bit strange only the first author of every paper is visible.

Summary:
The presented method solves an important problem, and the quantitative results seem reasonable. However, there’s limited novelty and a key design choice in using previously calculated deformable vector fields, is not justified.

**Paper Type:**

methodological development

**Strengths Weaknesses:**

Strengths:
The use case of registration uncertainty in quantifying uncertainty of dose delivery is well justified, and clearly important.
The validation against a variety of open source, and commercially available algorithms was good to see.

Weaknesses:
The reason why you would use a previously calculated mean is not really justified - this seems like quite an odd choice, it doesn’t necessarily affect the correctness of the work, but it needs to be explained.
The contributions of this work aren’t summarised in the intro.
For the b-spline based method - it seems quite counterintuitive to use a sparse parameterisation and then estimate voxelwise uncertainty.
The sparsity of the covariance means the derived samples are likely to have limited smoothness - however, we are not shown any example samples to evaluate this. It would be helpful to see some quantification of sample smoothness compared to the means.
The influence of regularisation on the posterior covariance is not discussed
The explanation in section 2.3 could be improved - writing the loss to discuss the likelihood and prior parts would simplify the explanation

---

### Official Review · Reviewer_r445 · 2022-02-21

**Rating:** 4
**Confidence:** 4
**Recommendation:** Short Oral

**Deanonymize Review:**

no

**Detailed Comments:**

the points to consider when revising/extending the paper:

- it wasn't straightway to notice that the displacement (coming from demons, bsplines, velocity) is also the input the network to predict the uncertainty. Potentially, this maybe better highlighted when introducing the methods . the description of the registration network is also limited, and make difficult to follow the paper

- Figure 3: are these 3 views from the same volume, or three different volumes (patients)

**Paper Type:**

both

**Strengths Weaknesses:**

the paper presents the method and the results for uncertainty estimation/prediction using the state-of-the-art deformable image registration (DIR) algorithms for the Adaptive Proton Therapy planning. the method is built based on a probabilistic VoxelMorph network, which additionally incorporates the traditional deformable image registration algorithms (e.g. demons, b-spline registration, Velocity). For the existing DIR algorithms, the network predicts only the uncertainty associated with the displacement resulting from the existing DIR algorithms.

Strengths:
 - Clear motivation, and clinical application when designing the registration uncertainty estimation. tracking daily changes in proton therapy is challenging yet unsolved problem and quantitative assessment of the uncertainty associated with image registration may add benefits for the personalized planning of the target margins

- an elegant combination of the existing methods for image registration merging both deep learning approach such as VoxelMorph, and the state-of-the-art optimization based algorithms (demons, bsplines)



- the paper is clearly written,

the weaknesses:
 - the methodological advances are limited - the method is based purely on the existing VoxelMorph framework.

- the results can be considered as a proof-of-concept, and while the results support the claims made in the paper, the method was not evaluated on the real proton therapy data, but rather on publicly available data for deformable image registration (DirLab not related to proton therapy). DirLab data set is a single breath cycle data, and thus not representative to the longitudinal changes expected in several weeks treatment. Furthermore, in both cases, deformable image registration  isn't necessary diffeomoprhic.

- motion in the lungs is quite complex to estimate (both 1) during the single breathing cycle due to sliding motion, 2) over several weeks, as registration may need to compensate for deformations related to treatment, that in turn exhibit significant anatomical changes), thus in future it may be interesting to include some more lung-specific image registration frameworks which model sliding motion [doi.org/10.1016/j.media.2014.05.005, doi.org/10.1016/j.media.2015.09.005]

---

### Decision · Program_Chairs · 2022-02-22

Accept